# Validation of Running Gait Event Detection Algorithms in a Semi-Uncontrolled Environment

**DOI:** 10.3390/s22093452

**Published:** 2022-04-30

**Authors:** Seth R. Donahue, Michael E. Hahn

**Affiliations:** Bowerman Sports Science Center, Department of Human Physiology, University of Oregon, Eugene, OR 97404, USA; sethd@uoregon.edu

**Keywords:** running, biomechanics, Inertial Measurement Unit, gait events, contact time

## Abstract

The development of lightweight portable sensors and algorithms for the identification of gait events at steady-state running speeds can be translated into the real-world environment. However, the output of these algorithms needs to be validated. The purpose of this study was to validate the identification of running gait events using data from Inertial Measurement Units (IMUs) in a semi-uncontrolled environment. Fifteen healthy runners were recruited for this study, with varied running experience and age. Force-sensing insoles measured normal foot-shoe forces and provided a standard for identification of gait events. Three IMUs were mounted to the participant, two bilaterally on the dorsal aspect of the foot and one clipped to the back of each participant’s waistband, approximating their sacrum. The identification of gait events from the foot-mounted IMU was more accurate than from the sacral-mounted IMU. At running speeds <3.57 m s^−1^, the sacral-mounted IMU identified contact duration as well as the foot-mounted IMU. However, at speeds >3.57 m s^−1^, the sacral-mounted IMU overestimated foot contact duration. This study demonstrates that at controlled paces over level ground, we can identify gait events and measure contact time across a range of running skill levels.

## 1. Introduction

The biomechanical analysis of running outside of the laboratory can be a useful tool for real-time or session-by-session feedback during training for coaches and athletes [1,2]. The standard in human locomotion research for identification of gait events is the use of three-dimensional (3D) motion capture and ground reaction force data. These systems typically utilize multiple cameras and force plates in a controlled laboratory environment. Despite being the gold standard, these methods require expensive equipment, large indoor facilities, and technical expertise [3,4,5], thus limiting their practical use in clinical or sporting environments. Previous studies have shown the efficacy of utilizing mobile sensing units for the detection of gait events inside and outside the laboratory, as the baseline information needed for biomechanical analysis [6,7]. However, there has been a lack of validation of these algorithms for gait event estimation outside the laboratory. This work expands upon current validation techniques for the estimation of gait events and foot contact duration during running in a semi-uncontrolled environment.

Over the past decade, there has been extensive work to understand Inertial Measurement Unit (IMU) data in the development of techniques to evaluate human locomotion and identify gait events for running in a controlled laboratory setting [8,9,10,11,12,13,14,15]. There are typically nine sensors in a standard IMU: tri-axial accelerometers (linear acceleration), tri-axial rate gyroscopes (angular velocity), and tri-axial magnetometers (magnetic field). These sensors have been used for the measurement of biomechanical variables in several environments and over different durations in and out of the laboratory [2,16,17]. However, data from IMUs cannot be used for analysis of biomechanical variables without thorough pre-processing of the data and specific algorithms [18]. These algorithms rely explicitly on expertise of the researcher to understand segmental accelerations and angular velocities throughout the gait cycle, and then develop rules for the estimation of foot contact, via gait events: initial contact (IC) and toe off (TO). Some of these approaches have been developed specifically for running, with sensors on the foot, shank, and the sacrum [11,12,13,19,20]. The application of these analyses have led to the measurement of trail running, marathons, and training runs for athletes ranging in skill from recreational to competitive collegiate athletes [8,16,21,22]. Prior work in this research space has shown that consistent features can be extracted from data in the laboratory and in the real-world, with modest efforts made to validate the estimated biomechanical outcomes in the real-world environment [23,24]. Force plates have been used previously to validate the identification of gait events [9,11], as have force-instrumented treadmills [17,25,26]. However, the validations of these methods did not account for the possible range of speeds and skill levels in an outdoor semi-uncontrolled environment.

The purpose of this work was to validate the identification of gait events using data from IMUs in a semi-uncontrolled environment. Second, this study sought to expand the range of speeds tested with these algorithms and expand the range of participant skill, from novice runners who run <5 miles per week to runners who can run a 5k race in a sub-15 min time. We expected the estimation of gait events and foot contact time to be more accurate for the foot-mounted IMUs in comparison to the sacral-mounted IMU across the range of speeds. Portable insoles developed for the measurement of in-shoe forces [27] were used as the standard for validation of the identification of IC and TO in a real-world environment. The algorithm developed in this study will be considered valid if the Root-Mean-Squared Error (RMSE) in overall contact time is less than 0.04 s across the range of speeds (representing <5–6% total contact time at jogging/running speeds).

## 2. Materials and Methods

Data were collected from 15 participants (9 male, 6 female, age: 23.6 ± 11.0 years, height: 178.3 ± 6.3 cm, mass: 73.5 ± 7.5 kg) as part of a larger study (Table 1). The study protocol was approved by the Institutional Review Board at the University of Oregon (protocol 10062020.007). Each participant provided written informed consent prior to enrollment in the study. All analyses were performed using custom Matlab programs (Mathworks, Natick, MA, USA). Multi-axis IMUs (Casio Computer Co., Ltd., Tokyo, Japan) were mounted on the dorsal aspect of the participants’ feet and approximately on the sacrum (clipped to the back of the participants waistband). These sensors recorded 3D linear accelerations and angular velocities at 200 Hz. Inertial data were postprocessed with a Kalman filter to orient the vertical axis of the local (IMU) coordinate system to gravity, for both foot- and sacral-mounted IMUs. Foot-shoe normal force data were recorded with Loadsol insole force sensors (Novel Electronics, St. Paul, MN, USA) at 100 Hz. Standard GPS data were measured with a Garmin Forerunner (Garmin, Olathe, KS, USA). Participants performed progressively faster 400 m running trials (four to five, with the fastest speed being optional) on a square practice track, based on the self-reported race pace for a 5 km event. An example of the paces run by a participant is shown in Table 2. The total range of speeds run by participants was 2.4–5.4 m s^−1^. The participants monitored their lap-time with a standard wrist-mounted Garmin GPS display. They were asked to complete a 400 m trial within two seconds of the given time corresponding to an average pace. If the participant missed this time by more than 2 s, they were asked to repeat that pace after suitable rest. These speeds represent typical training and race paces for the majority of recreational and high-level distance runners [8,28].

Foot-shoe normal force data measured from force-sensing insoles were considered the standard reference for identification of measured gait events [27]. Inertial signals and force data were time-synced using controlled unilateral ‘foot-stomps’ before and after each trial. The IMU data were then down-sampled to match the force-sensing insole sampling frequency (100 Hz) and filtered using a 4th order low-pass zero-lag Butterworth filter (*fc* = 35 Hz) (Figure 1). This filter was chosen as it was more conservative in reduction in noise for the accurate identification of gait events, using peak accelerations [8]. Force data <50 N were set to zero.

Algorithms used in this study are briefly described here. A more thorough treatment of the algorithms can be found in Figure 1. The identification of gait events with foot-shoe normal force data utilized a threshold of 50 N. Gait event estimation from the IMU data utilized distinct spatial and temporal rules. The identification of gait events from the dorsal-mounted IMUs estimated initial contact by identifying peaks in the resultant acceleration. The spatial rule for initial contact with the foot-mounted IMUs (IC_foot_) was a minimum resultant acceleration of 50 m s^−2^. The temporal rule for determining IC_foot_ was a minimum duration of 500 ms between estimated consecutive IC_foot_ [29]. The identification of toe off from the foot-mounted IMUs (TO_foot_) was performed by searching a specific temporal window beginning 100 ms after the estimated IC_foot_, ending at the half-width of the estimated stride time. In this window, TO_foot_ was either identified as the local maxima of vertical acceleration or the first instance when the vertical acceleration was greater than three times gravity [13,29]. Gait event detection using the sacral IMU utilized the anterior posterior accelerations. The spatial rule for the identification of initial contact from the sacral-mounted IMU (IC_sacrum_) was local minima with a maximum value of 5 m s^−2^ in the posterior direction. The temporal rule for IC_sacrum_ was a minimum temporal difference of 200 ms between the identified IC_sacrum_ [29]. The identification of toe off from the sacral-mounted IMU (TO_sacrum_) with a search window was either with the maximum acceleration in the anterior direction or the maximum positive slope of the acceleration in the anterior direction [9]. Exemplar outputs of these algorithms are presented in Figure 2 and Figure 3.

## 3. Results

The gait events estimated using data from foot-mounted IMUs were more accurate than those estimated using the sacral-mounted IMU when compared to the force measurement standard identification (Figure 4, Table 3). Foot-mounted IMUs had a larger RMSE than the sacral-mounted IMU for the identification of gait events in the slowest running speed conditions. Across the range of speeds, the algorithms identified IC_foot_ and IC_sacrum_ with similar accuracy (Figure 4, Table 3). The identification of TO_sacrum_ was more accurate at slower speeds than TO_foot_. However, at running speeds >3.57 m s^−1^, the algorithm identified TO_sacrum_ after the force-measured TO and with larger RMSEs than TO_foot_ (Figure 4).

Estimation of the foot contact duration with data from the foot-mounted IMU showed an overestimation of foot contact duration at slower running speeds (<2.55 m s^−1^). The estimated foot contact duration from the dorsal-mounted IMUs generally had a smaller RMSE across the range of speeds than the sacral IMU-estimated contact duration (Table 3 and Figure 5). The algorithm for the sacral-mounted IMU data consistently underestimated the duration of foot contact at slower speeds, <3.5 m s^−1^, and overestimated the duration of foot contact at faster speeds, >3.5 m s^−1^ (Figure 5).

Analysis of foot contact by trial mean was examined using Bland–Altman plots. These plots present a comparison between the average difference between IMU-estimated foot contacts and force-measured foot contacts (Figure 6). The offset of the foot IMU estimate was 0.004 s with [−0.005 0.013] 95% Limits of Agreement (LoA) and the sacral IMU estimate offset was 0.001 s with [−0.018 0.021] 95% LoA. These results show more variability at the slower speeds, and longer foot contacts, for both the sacral- and foot-mounted IMUs (Figure 6). We used a linear model to examine the relationship between the IMU-estimated foot contacts and the force-measured foot contact as well (Figure 7). Regression analysis of the sacral estimation of foot contact resulted in an r^2^ value of 0.73, a moderate correlation, and a slope of 0.60, indicating an underestimation of the foot contact duration. Regression analysis from the foot IMU-estimated contact duration resulted in an r^2^ value of 0.91, a strong correlation, and a slope of 1.15, indicating a slight overestimation of the foot contact duration.

## 4. Discussion

The purpose of this work was to validate the identification of gait events using data from IMUs in a semi-uncontrolled environment. We collected inertial data from two IMUs attached bilaterally on each foot and one approximately on the sacrum, from participants of varying running skill levels. We developed and implemented two algorithms for the identification of gait events from an IMU, based upon previous work [13,20,29]. The outputs of these algorithms, the IMU-estimated gait events, were validated against the standard of a force-sensing insole. The main findings of the work are summarized briefly here: (1) estimated and measured contact times generally decreased across the range of running speeds; (2) the identification of gait events from the foot-mounted IMUs was more accurate than the identification of gait events from the sacral-mounted IMU; (3) foot contact was identified with an average RMSE of <0.04 s across the range of average running speeds for the foot- and sacral-mounted IMUs.

It is necessary to address the accuracy of the force-sensing insoles and their measurement of contact time with respect to speed, as this measure was considered our standard. Foot contact time measured from the insoles followed a similar pattern to [28]: a decrease in contact time with an increase in speed [30]. Foot contact durations as measured by the force insoles were consistently longer in duration than those reported in [28]. A contributing factor in this was the participants recruited for these studies. Our study recruited participants from a range of running skills, from the truly novice to highly trained community runners. The samples tested in previous studies consisted of middle distance and sprint athletes.

The time between IC_foot_ and the force-sensing insole approached zero difference as the speed increased (Figure 5). We used an algorithmic method most similar to [9], which proposed the identification of peak resultant acceleration across the range of speeds, and this has become a common approach for the identification of IC_foot_ [9,31]. The work of [9] utilized a force plate and single foot strikes for the identification of gait events, reporting differences in IC_foot_ ranging from −7.3 to 3.3 ms. In our study, the error in the estimate of IC_foot_ ranged from −63 to −5 ms (Figure 4). They additionally reported TO differences ranging from −53 ms to −32 ms, compared to our range of TO_foot_ differences from −78 to 2 ms (Figure 4) [9]. We improved on the identification of TO_foot_, by incorporating their rule for the identification of TO_foot_ as a secondary rule to the identification of peak vertical acceleration, in a temporal search window [9,29]. Our work had more variability in the range of TO_foot_ identification only at the three slowest running speeds, however, as there was only one participant who ran at these average running speeds.

Foot-mounted IMUs overestimated the contact time at running speeds <2.52 m s^−1^ (Figure 4 and Figure 6). The slope of the regression line for this comparison was 1.15, providing further evidence of the overestimation of foot contact time overall (Figure 7). Another study reported an offset of −0.047 s with a 95% LoA of [−0.059 0.154] [29], compared with our findings of 0.004 s with a 95% LoA [−0.005 0.013], showing an overall improvement. It should be noted that Benson et al. reported challenges in the identification of TO_foot_ for one of their participants, which may have contributed to the larger offset [29]. Our algorithm was developed on a wider range of speeds, participant running abilities, and on data collected in a semi-uncontrolled environment. The set of algorithms we developed captured the gait events more effectively than previous work. The overestimation of foot contact at the slower running speeds would likely be remedied by the inclusion of a greater number of less experienced runners.

Estimations of contact time from the sacral-mounted IMUs between the speeds of 2.52 and 3.16 m s^−1^ matched the measured contact times (Figure 5). However, the RMSE TO_sacrum_ at speeds >3.57 m s^−1^ was greater than 0.04 (Table 3). This stems from the difficulty in estimating the temporal window in which to identify TO_sacrum_. The identification of the temporal duration of the window in which to estimate TO_sacrum_ is related to aerial time, which, in this study, ranged from 40 to 100 ms. Differing window lengths were tested to accurately identify TO_sacrum_. The most accurate of these resulted from window termination 20 ms before the next IC_sacrum_ (Figure 3). A dynamic temporal window for the estimation of TO_sacrum_ could be a way to further improve the estimation accuracy of TO_sacrum_ and foot contact. The study by [26] reported an average underestimation of foot contact from sacral-mounted IMUs from −0.017 to −0.001 s, while the current findings showed average contact time differences from −0.011 to 0.027 s. Specifically, at average running speeds <2.56 m s^−1^, the sacral IMU underestimated foot contact time, and at average running speeds >3.16 m s^−1^, the contact time was overestimated. Another study [29] reported a foot contact offset from a sacral-mounted IMU of 0.029 s with a 95% LoA [−0.069 0.010]. Our analysis had a smaller offset of 0.001 s with a 95% LoA of [−0.018 0.021].

There were multiple limitations in this study. First, the temporal synchronization between the IMUs and the force-sensing insoles presented an initial challenge. While participants were running, the IMU clock and the force-sensing insole clock could be off by 0.01 s, which thus accumulates the longer the measures are taken. These errors resulted in zeros being added or removed during the swing of the phase from the force-sensing insole data. We partially accounted for time drift using the time synchronization between each trial. However, we did not want to artificially decrease the RMSE in the results of this work by removing data due to imperfect synchronization. This, in turn, led to cumulative errors between the measured and estimated gait events. These errors would be more concerning if there were large differences in the estimation of contact time across the range of speeds. As it is, we feel that it is important to include the data in full, to fully represent the performance of the techniques used.

A single participant’s data heavily influenced the error in the model. This participant ran less than 5 miles per week and was truly a novice runner. When the data from this participant were removed from the dataset, IC_foot_ ranged from −20 ms to −1 ms, compared to the original −63 ms to 5 ms (a ~60% improvement) from speeds of 2.67 m s^−1^ to 5.4 m s^−1^, and TO_foot_ temporal differences ranged from -30 ms to -1 ms, compared to the original values of −78 ms to 2 ms (a ~75% improvement) across the range of speeds. Further, the exclusion of this participant from the analysis reduced the offset in the foot-mounted IMU in the estimation of foot contact from an offset of 4 ms to an offset of 1 ms, while the 95% LoA remained the same. The slope of the linear regression also decreased from 1.15 to 1.10 when this participant’s data were removed from analysis, indicating a reduced overestimation of contact time in the model. We chose to include this participant as a representative example of the truly novice runner. We expect that if more runners from a wide range of running levels were included in the dataset, we would see decreased variability in the identification of IC and TO from the minimal and maximal running speeds included in this work.

## 5. Conclusions

In conclusion, our results demonstrate the validity of two different gait event detection algorithms for a range of running speeds and skill levels in a semi-uncontrolled environment. We used data from a wider range of participant skill levels, and a wider range of running speeds than previous studies. We demonstrated the utility of these algorithms for the identification of foot contacts in a semi-uncontrolled environment. The use of a gait event detection system in a real-world environment needs to be validated for a broader set of conditions before we can estimate other biomechanical variables from these devices. The next steps in this research are the estimation of ground reaction force waveforms from the force-sensing insole data, and testing of these algorithms in a truly uncontrolled environment.

## Figures and Tables

**Figure 1 sensors-22-03452-f001:**
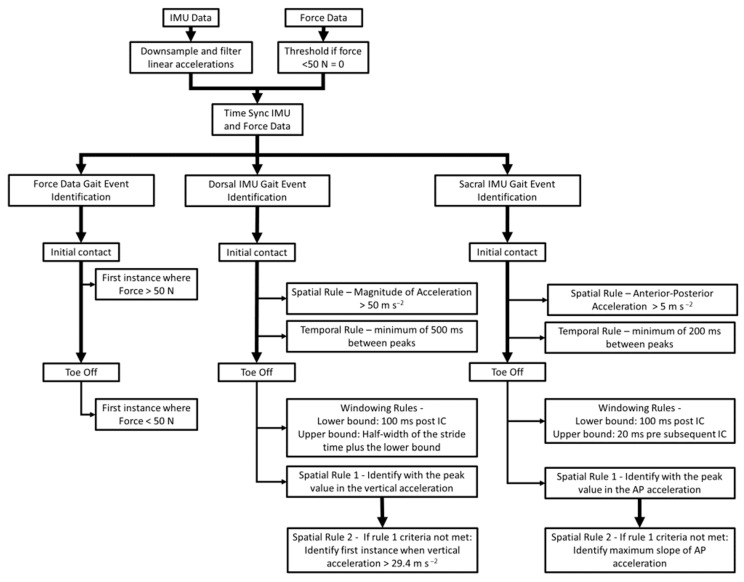
Flow chart of the data postprocessing steps and the algorithmic methods for the identification of gait events from the force-sensing insole data, and data from multi-axial IMUs at both anatomical locations.

**Figure 2 sensors-22-03452-f002:**
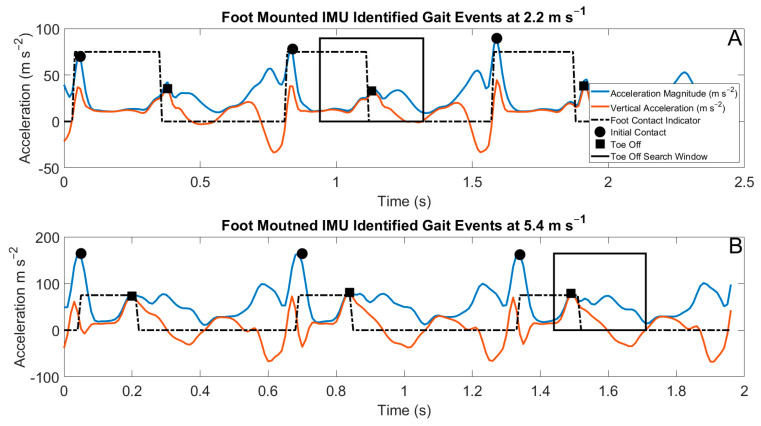
Acceleration waveforms from the IMUs mounted on the right foot (orange and blue waveforms), with superimposed foot contacts (dashed black square waves) identified from the force-measuring insoles. The estimated IC_foot_ is shown in the filled circles, and the estimated TO_foot_ is shown in the filled squares. The search windows used for the identification of TO_foot_ are shown in the solid black rectangles. (**A**) Data from a 2.24 m s^−1^ run; (**B**) data from a 5.36 m s^−1^ run.

**Figure 3 sensors-22-03452-f003:**
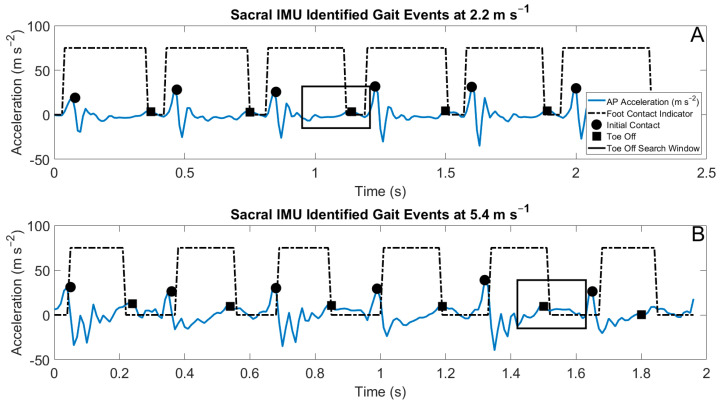
Acceleration waveforms from IMUs mounted on the sacrum (blue waveforms), with superimposed foot contacts (dashed black square waves) identified from the force-measuring insoles. The estimated IC_sacrum_ is shown in the filled circles and the estimated TO_sacrum_ is shown in the filled squares. The search windows used for the identification of TO_sacrum_ are shown in the solid black rectangles. (**A**) Data from a 2.24 m s^−1^ run; (**B**) data from a 5.36 m s^−1^ run.

**Figure 4 sensors-22-03452-f004:**
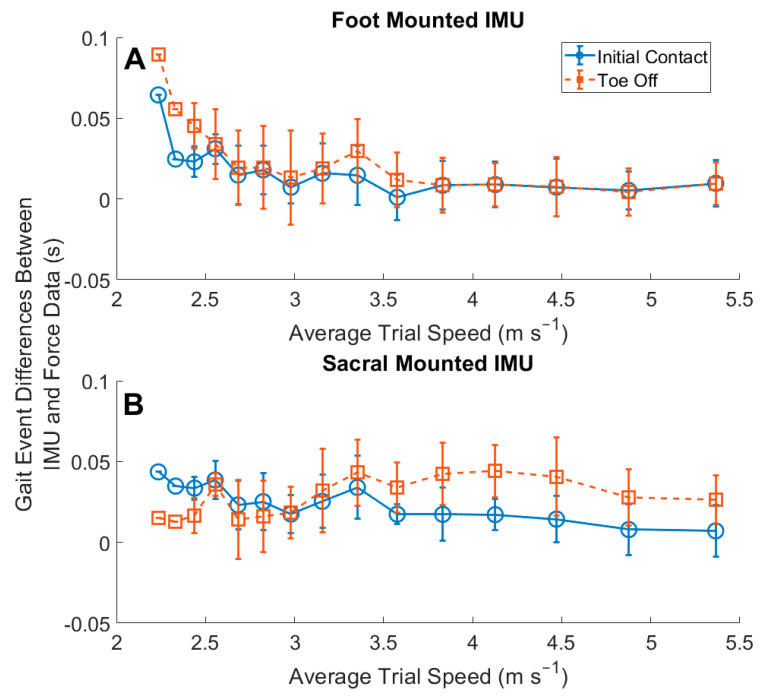
Time differences in the identification of gait events between measured forces and estimated IMU gait events: (**A**) results from foot-mounted IMU; (**B**) results from sacral-mounted IMU. Negative time differences indicate that the IMU-estimated gait event occurred prior to the measured gait event. The identification of TO_sacrum_ had larger error rates due to the temporal windowing of the data, and the wider range of speeds used than previous work.

**Figure 5 sensors-22-03452-f005:**
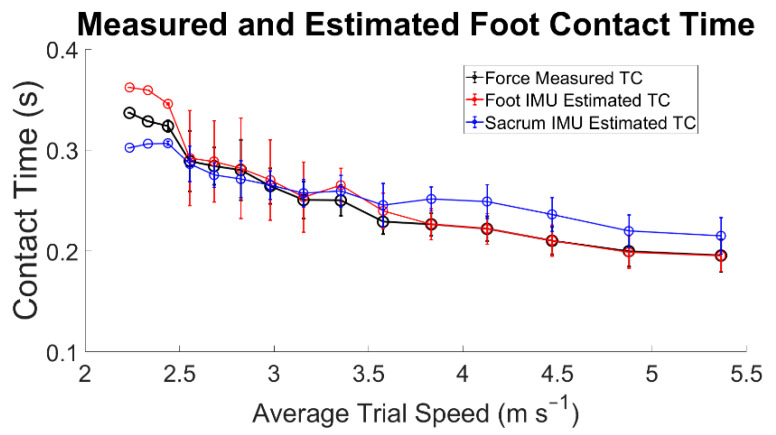
Differences in force-measured and IMU-estimated contact time across the range of average speed trials. The sample sizes for each speed are shown in Table 1.

**Figure 6 sensors-22-03452-f006:**
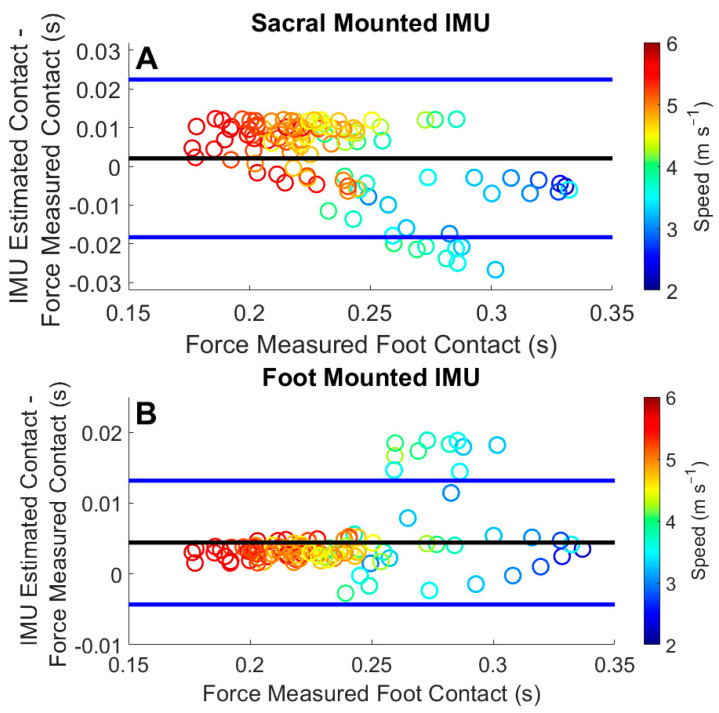
Bland–Altman plot displaying the average differences between the estimated contact durations from the IMUs and the measured gait events from the force-sensing insoles: (**A**) data from the sacral-mounted IMU; (**B**) data from the foot-mounted IMU. Each dot represents an average speed trial by a participant. Differences greater than 0 are an overestimation of contact time by the IMU. Differences less than 0 are an underestimation of foot contact by the IMU.

**Figure 7 sensors-22-03452-f007:**
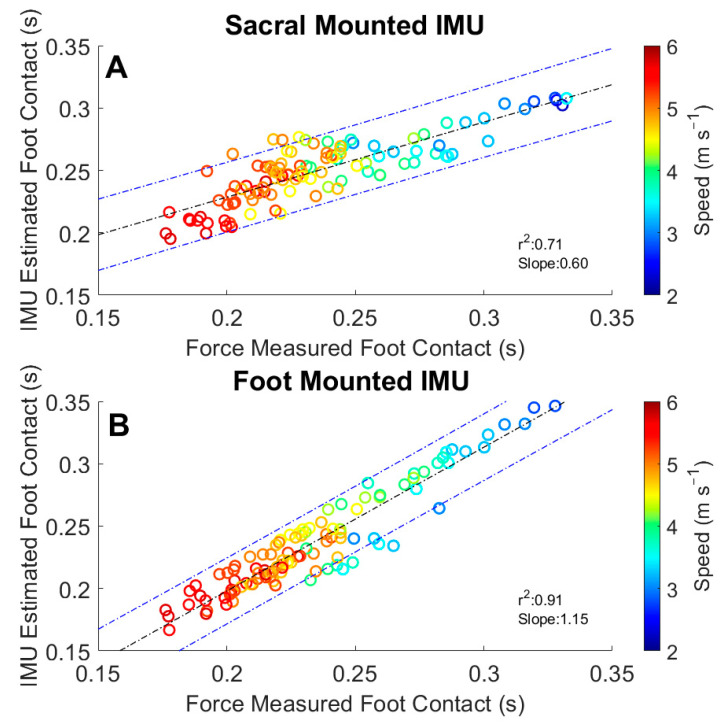
Time differences between the IMU-estimated foot contact and force-measured foot contact: (**A**) data from sacral-mounted IMU; (**B**) data from foot-mounted IMU. Sacral-mounted IMU data generally underestimated foot contact, with a slope of 0.60. Foot-mounted IMU data generally overestimated foot contact duration at a faster speeds slope of 1.15. The foot-mounted IMU was more accurate across the range of speeds for the estimation of foot contact in comparison to the sacral-mounted IMU.

**Table 1 sensors-22-03452-t001:** Distribution of average running speeds.

Average Running Velocity (m s^−1^)	Number of Participants
2.23	1
2.33	1
2.43	1
2.55	4
2.68	6
2.82	6
2.98	6
3.15	7
3.35	9
3.57	9
3.83	9
4.12	8
4.47	8
4.87	7
5.36	7

**Table 2 sensors-22-03452-t002:** Exemplar paces for participant with a 7 min per mile, 5 km race pace, and expected average velocity-corresponding pace in minutes per mile.

Example Paces	Average Velocity (m s^−1^)	Minutes per Mile
Pace 1	3.15	8:30
Pace 2	3.35	8:00
Pace 3	3.57	7:30
Pace 4	3.83	7:00
Pace 5 (optional)	4.12	6:30

**Table 3 sensors-22-03452-t003:** Root-mean-square error for identification of gait events and estimation of contact time.

	Foot-Mounted			Sacral-Mounted		
Velocity (m s^−1^)	Initial Contact (s)	Toe Off (s)	Contact Time (s)	Initial Contact (s)	Toe Off (s)	Contact Time (s)
2.24	0.074	0.092	0.025	0.045	0.059	0.043
2.33	0.026	0.060	0.045	0.038	0.021	0.030
2.44	0.024	0.044	0.026	0.035	0.027	0.026
2.55	0.033	0.064	0.054	0.039	0.036	0.024
2.68	0.024	0.040	0.038	0.028	0.028	0.490
2.82	0.024	0.034	0.029	0.030	0.032	0.030
2.98	0.018	0.039	0.035	0.024	0.032	0.027
3.16	0.024	0.036	0.032	0.029	0.042	0.029
3.35	0.024	0.035	0.025	0.039	0.051	0.036
3.58	0.018	0.023	0.022	0.019	0.042	0.037
3.83	0.020	0.031	0.028	0.024	0.045	0.036
4.13	0.019	0.024	0.023	0.020	0.047	0.039
4.47	0.021	0.026	0.021	0.020	0.044	0.036
4.88	0.018	0.026	0.022	0.020	0.035	0.030
5.36	0.020	0.038	0.035	0.020	0.033	0.026

## Data Availability

The data presented in this study are available on request from the corresponding author. The data are not publicly available, due to the terms of an industry-sponsored research agreement.

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
