# Peer review of "Validation of Running Gait Event Detection Algorithms in a Semi-Uncontrolled Environment"

_sensors, 2022, doi:10.3390/s22093452_

Round 1
Reviewer 1 Report
Use of wearable sensors to monitor human running has attracted great attention as portable gait measurement methods, and the first important step is to identify key gait events: heel contact and toe-off. Overall, the current study has reported successful results on IMU-based gait detection methods. The study is straightforward and provides useful insights on this topic. It is, however, important to properly describe how IMUs were attached to the foot and sacrum because orientation and position were, arguably, the most important determinants when applying IMUs. Further comments are provided below.
General
Please provide where and how the IMUs were attached to the foot and sacrum (i.e. position and orientation). It is probably most helpful if an illustration or photo is provided to clarify how they were used. In addition to data processing methods, the position and orientation are the critical determinants for gait event detection.
Specific
- Line 62, 85, 215 etc: What is 5k? Is it different from 5km? Be consistent with abbreviation.
- Line 86-88: ‘The participants monitored their pace with a standard wrist-mounted Garmin GPS display. If a participant missed their pace by more than 2 seconds, they were asked to repeated that pace after suitable rest.’ This part is not easy to understand. What is the pace? Is it like an auditory beeping that tells the timing of each stepping? Is it easy to keep the pace?
- Line 104-: When using IC for the first time, please spell out what it means. Same applies to TO.
- Table 2: The description of ‘Minutes per Mile’ is a bit confusing. It may be better to stick to the metric system.
- Line 215: ‘…run at 15 min 5 km race pace’ I am not too sure what it exactly means but people who can run 5 km in 15 minutes are considered to be at the highly elite level. Is this the case?
- Line 245: ‘in which to identify TOscrum’ Please fix the errors.
Author Response
Comments and Suggestions for Authors
Use of wearable sensors to monitor human running has attracted great attention as portable gait measurement methods, and the first important step is to identify key gait events: heel contact and toe-off. Overall, the current study has reported successful results on IMU-based gait detection methods. The study is straightforward and provides useful insights on this topic. It is, however, important to properly describe how IMUs were attached to the foot and sacrum because orientation and position were, arguably, the most important determinants when applying IMUs. Further comments are provided below.
General
Please provide where and how the IMUs were attached to the foot and sacrum (i.e. position and orientation). It is probably most helpful if an illustration or photo is provided to clarify how they were used. In addition to data processing methods, the position and orientation are the critical determinants for gait event detection.
Response: Thank you very much for your feedback. The sensors are clipped to the shoes and on the back of the participant’s waistband. The IMUs have clips built onto them. The methodology the orientation of each of the sensor’s orientation is corrected to gravity, i.e. the vertical direction of the IMU is oriented to gravity. During the periods of when the angular velocity is approximately zero and the total magnitude of the accelerations are approximately 9.8 m s-2 the Kalman filter resets the orientation of the sensor coordinate system to gravity. This occurs during stance phase for the foot mounted IMUs and during the aerial phase for the sacral mounted IMU. These details are provided in the Methods section of the revised manuscript.
Specific
- Line 62, 85, 215 etc: What is 5km? Is it different from 5km? Be consistent with abbreviation.
Response: Thank you for the clarifying feedback, we have adjusted these for consistency.
- Line 86-88: ‘The participants monitored their pace with a standard wrist-mounted Garmin GPS display. If a participant missed their pace by more than 2 seconds, they were asked to repeated that pace after suitable rest.’ This part is not easy to understand. What is the pace? Is it like an auditory beeping that tells the timing of each stepping? Is it easy to keep the pace?
Response: Thank you for the comment. We have revised the wording in this section to better reflect the protocol used.
- Line 104-: When using IC for the first time, please spell out what it means. Same applies to TO.
Response: Thank you, we have clarified the abbreviation of IC and TO specifically for the foot and the sacrum in the methods section of the paper.
- Table 2: The description of ‘Minutes per Mile’ is a bit confusing. It may be better to stick to the metric system.
Response: We have added a column to Table 2 with the metric value in meters per second and the corresponding minutes per mile pace. The reason for including both units is to better reach audiences who use either unit for tracking the running pace.
- Line 215: ‘…run at 15 min 5 km race pace’ I am not too sure what it exactly means but people who can run 5 km in 15 minutes are considered to be at the highly elite level. Is this the case?
Response: Thank you for the feedback here, we have adjusted the wording to clarify that these are highly trained community runners.
- Line 245: ‘in which to identify TOscrum’ Please fix the errors.
Response: Thank you for bringing this to our attention. We have fixed these typos.
Reviewer 2 Report
Introduction
- Use the abbreviation RMSE and mention in full in the first time you use it;
Materials and methods
- What is the study design?
- Why did the authors use only 15 volunteers?
- Was there a calculation to delimit the sample size? If not, it should be mentioned as another limitation of the study;
- Was there an assessment of the types of feet of the volunteers? This could influence the results of the data.
- Feet with pronated and supinated footsteps may present greater pressure on the medial and lateral edges of the feet, respectively. Therefore, I believe that the types of feet and tread of the volunteers should be mentioned by the authors.
- In addition to the type of foot and tread, gait and running have other factors that influence their performance, such as body posture, and problems in the sensory systems (visual, vestibular and somatosensory) responsible for regulating body balance and influencing gait directly and are not mentioned by the authors, these issues may influence the results of the study and should be mentioned in the paper.
- The standard deviation of the sample variables mentioned in the methods was not mentioned, it must be included by the authors, or its confidence interval;
Discussion
When reading this article, I expected a more clinical discussion, bringing the elements of validation to its clinical application in athlete-runners, which did not happen. The authors focused more on the discussion of the validation findings only, however, I believe that the validation of these algorithms has a clinical impact on the performance of these athlete-runners, so I suggest adding one or more paragraphs to the discussion on how would be the use or application of this instrument in the clinical practice of the population in question.
Author Response
Comments and Suggestions for Authors
Introduction
- Use the abbreviation RMSE and mention in full in the first time you use it;
Response: Thank you for the feedback. We have made this revision.
Materials and methods
- What is the study design?
Response: There was no specific study design per se. This work is more aptly categorized as method development, specifically development of algorithms for biomechanical analysis. We have attempted to make this clear throughout the work.
- Why did the authors use only 15 volunteers?
Response: This is an interesting question, thank you. This study used convenience sampling, as it was difficult to recruit participants throughout the Covid-19 pandemic. Throughout this process we worked to recruit a variety of participants, through means of social media, and word of mouth during periods of quarantine. The sample size is similar to previously published work in this space, see Clermont et al. 2020.
- Was there a calculation to delimit the sample size? If not, it should be mentioned as another limitation of the study;
Response: Thank you for this question. As this work is more focused on developing a model and is not testing a specific hypothesis, we did not conduct a a sample size estimation for any specific statistical power.
-Was there an assessment of the types of feet of the volunteers? This could influence the results of the data. Active runners of no reported problems, of excessive eversion, generally neutral feet. Not have orthopedic conditions including foot morphology as normal. There were no assessments of the participants feet…
Response: This is an interesting question. None of the participants reported any foot abnormalities in foot morphology. We are not able to assess foot morphology using the Loadsol force sensing insole, as this device records a single, summative normal force, rather than distributed force over a set of pressure-measuring sensors. Future work may examine these differences; however, this was not the purpose of the current study. Foot morphology may be important in running injury, however for the identification of gait events from inertial data with a sample of able-bodied participants, with no reported abnormalities we did not make any further assessment of foot type or search for abnormal foot morphology.
- Feet with pronated and supinated footsteps may present greater pressure on the medial and lateral edges of the feet, respectively. Therefore, I believe that the types of feet and tread of the volunteers should be mentioned by the authors.
Response: The participants recruited for this study were normal able-bodied participants with no reported or evident running injuries. As mentioned above, we were not able to measure whether force was exerted more to the medial or lateral side of the foot, as the force sensing insoles used in this study measure only normal ground reaction forces from the whole sensor, unlike pressure-mapping insoles. Such an approach would be appropriate for a more clinically oriented manuscript.
- In addition to the type of foot and tread, gait and running have other factors that influence their performance, such as body posture, and problems in the sensory systems (visual, vestibular and somatosensory) responsible for regulating body balance and influencing gait directly and are not mentioned by the authors, these issues may influence the results of the study and should be mentioned in the paper.
Response: Thank you for the inquiry. The overarching purpose of this study is the development of algorithms for identification of gait events outside of the laboratory. This was purely an observational study. It was assumed that everyone must have adequate balance control in order to train to the level necessary to meet the inclusion criteria of the study. Our dataset does not contain the information required to test the suggested research questions.
- The standard deviation of the sample variables mentioned in the methods was not mentioned, it must be included by the authors, or its confidence interval;
Response: Thank you for identifying this missing information. These standard deviations have been added.
Discussion
When reading this article, I expected a more clinical discussion, bringing the elements of validation to its clinical application in athlete-runners, which did not happen. The authors focused more on the discussion of the validation findings only, however, I believe that the validation of these algorithms has a clinical impact on the performance of these athlete-runners, so I suggest adding one or more paragraphs to the discussion on how would be the use or application of this instrument in the clinical practice of the population in question.
Response: Thank you very much for the feedback. These algorithms have the potential to have immense clinical impact. However, we first need to validate these measures against the gold standard of the treadmill before we can make guesses at the functionality of these applications for clinical interpretation. We need to be cautious about what these sensors can and cannot measure, and the sensitivity of these sensors to different clinical measurements. Future work will include analysis of data collected from similar samples of runners on a force-instrumented treadmill while also wearing the force sensing insoles, as well as using IMUs and 3D motion capture. Analysis of those data will allow a more full validation of these novel models, which will then lead to more opportunities for clinical application.
Round 2
Reviewer 2 Report
Thank you for answering the questions, it made the reading of the text clearer.